# Plasma Sphingomyelin Disturbances: Unveiling Its Dual Role as a Crucial Immunopathological Factor and a Severity Prognostic Biomarker in COVID-19

**DOI:** 10.3390/cells12151938

**Published:** 2023-07-26

**Authors:** Diana Mota Toro, Pedro V. da Silva-Neto, Jonatan C. S. de Carvalho, Carlos A. Fuzo, Malena M. Pérez, Vinícius E. Pimentel, Thais F. C. Fraga-Silva, Camilla N. S. Oliveira, Glaucia R. Caruso, Adriana F. L. Vilela, Pedro Nobre-Azevedo, Thiago V. Defelippo-Felippe, Jamille G. M. Argolo, Augusto M. Degiovani, Fátima M. Ostini, Marley R. Feitosa, Rogerio S. Parra, Fernando C. Vilar, Gilberto G. Gaspar, José J. R. da Rocha, Omar Feres, Gabriel P. Costa, Sandra R. C. Maruyama, Elisa M. S. Russo, Ana Paula M. Fernandes, Isabel K. F. M. Santos, Adriana Malheiro, Ruxana T. Sadikot, Vânia L. D. Bonato, Cristina R. B. Cardoso, Marcelo Dias-Baruffi, Átila A. Trapé, Lúcia H. Faccioli, Carlos A. Sorgi

**Affiliations:** 1Department of Clinical, Toxicological and Bromatological Analysis, Faculty of Pharmaceutical Sciences of Ribeirão Preto–FCFRP, University of Sao Paulo–USP, Ribeirão Preto 14040-903, SP, Brazil; dianamota.t@gmail.com (D.M.T.); pedrovieira_sn11@hotmail.com (P.V.d.S.-N.); jonatancarvalho@usp.br (J.C.S.d.C.); cafuzo@usp.br (C.A.F.); malenac@usp.br (M.M.P.); viniciuspimentel@usp.br (V.E.P.); camilla.narjara@outlook.com (C.N.S.O.); g_rigotto@hotmail.com (G.R.C.); elisa@fcfrp.usp.br (E.M.S.R.); cristina@fcfrp.usp.br (C.R.B.C.); mdbaruff@fcfrp.usp.br (M.D.-B.); faccioli@fcfrp.usp.br (L.H.F.); 2Postgraduate Program in Basic and Applied Immunology–PPGIBA, Institute of Biological Sciences, Federal University of Amazonas–UFAM, Manaus 69080-900, AM, Brazil; malheiroadriana@yahoo.com.br; 3Department of Chemistry, Faculty of Philosophy, Sciences and Letters of Ribeirão Preto–FFCLRP, University of São Paulo–USP, Ribeirão Preto 14040-901, SP, Brazil; avilela@usp.br (A.F.L.V.); pedronobre@usp.br (P.N.-A.); thiago.felippe@usp.br (T.V.D.-F.); 4Department of Biochemistry and Immunology, Faculty of Medicine of Ribeirão Preto–FMRP, University of São Paulo–USP, Ribeirão Preto 14049-900, SP, Brazil; thaisfragasilva@gmail.com (T.F.C.F.-S.); imsantos@fmrp.usp.br (I.K.F.M.S.); vlbonato@fmrp.usp.br (V.L.D.B.); 5Department of General and Specialized Nursing, School of Nursing of Ribeirão Preto–EERP, University of São Paulo–USP, Ribeirão Preto 14040-902, SP, Brazil; jamilleargolo@usp.br (J.G.M.A.); anapaula@eerp.usp.br (A.P.M.F.); 6Hospital Santa Casa de Misericórdia de Ribeirão Preto, Ribeirão Preto 14085-000, SP, Brazil; augustomd@msn.com (A.M.D.); tata_ostini@hotmail.com (F.M.O.); 7Department of Surgery and Anatomy, Faculty of Medicine of Ribeirão Preto-FMRP, University of São Paulo–USP, Ribeirão Preto 14049-900, SP, Brazil; mrfeitosa@hcrp.usp.br (M.R.F.); rsparra@hcrp.usp.br (R.S.P.); jjrocha1@bol.com.br (J.J.R.d.R.); omar.feres@hspaulo.com.br (O.F.); 8Hospital São Paulo, Ribeirão Preto 14025-100, SP, Brazil; fcvilar@gmail.com; 9Department of Internal Medicine, Faculty of Medicine of Ribeirão Preto–FMRP, University of São Paulo–USP, Ribeirão Preto 14049-900, SP, Brazil; ggaspar@hcrp.usp.br; 10School of Physical Education and Sport of Ribeirão Preto, University of São Paulo–USP, Ribeirão Preto 14040-900, SP, Brazil; costa.gabriel@usp.br (G.P.C.); atrape@usp.br (Á.A.T.); 11Department of Genetics and Evolution, Center for Biological and Health Sciences, Federal University of São Carlos (UFSCar), São Carlos 13565-905, SP, Brazil; sandrarcm@ufscar.br; 12Department of Internal Medicine, Division of Pulmonary, Critical Care and Sleep, College of Medicine, University of Nebraska Medical Center, Omaha, NE 68198, USA; ruxana.sadikot@emory.edu

**Keywords:** sphingolipids, sphingomyelin, biomarker, COVID-19, inflammation

## Abstract

SARS-CoV-2 infection triggers distinct patterns of disease development characterized by significant alterations in host regulatory responses. Severe cases exhibit profound lung inflammation and systemic repercussions. Remarkably, critically ill patients display a “lipid storm”, influencing the inflammatory process and tissue damage. Sphingolipids (SLs) play pivotal roles in various cellular and tissue processes, including inflammation, metabolic disorders, and cancer. In this study, we employed high-resolution mass spectrometry to investigate SL metabolism in plasma samples obtained from control subjects (*n* = 55), COVID-19 patients (*n* = 204), and convalescent individuals (*n* = 77). These data were correlated with inflammatory parameters associated with the clinical severity of COVID-19. Additionally, we utilized RNAseq analysis to examine the gene expression of enzymes involved in the SL pathway. Our analysis revealed the presence of thirty-eight SL species from seven families in the plasma of study participants. The most profound alterations in the SL species profile were observed in patients with severe disease. Notably, a predominant sphingomyelin (SM d18:1) species emerged as a potential biomarker for COVID-19 severity, showing decreased levels in the plasma of convalescent individuals. Elevated SM levels were positively correlated with age, hospitalization duration, clinical score, and neutrophil count, as well as the production of IL-6 and IL-8. Intriguingly, we identified a putative protective effect against disease severity mediated by SM (d18:1/24:0), while ceramide (Cer) species (d18:1/24:1) and (d18:1/24:0)were associated with increased risk. Moreover, we observed the enhanced expression of key enzymes involved in the SL pathway in blood cells from severe COVID-19 patients, suggesting a primary flow towards Cer generation in tandem with SM synthesis. These findings underscore the potential of SM as a prognostic biomarker for COVID-19 and highlight promising pharmacological targets. By targeting sphingolipid pathways, novel therapeutic strategies may emerge to mitigate the severity of COVID-19 and improve patient outcomes.

## 1. Introduction

The ongoing global pandemic known as the 2019 coronavirus disease (COVID-19) is caused by severe acute respiratory syndrome coronavirus 2 (SARS-CoV-2). While the majority of patients with mild or moderate symptoms have a positive prognosis [1], there are cases where individuals progress to a severe or critical phase, characterized by severe respiratory failure requiring mechanical ventilation and intensive care unit hospitalization [2]. Moreover, this infection can have an impact on various organ systems, such as the neurological, cardiovascular, gastrointestinal, renal, hematological, and immunological systems [3]. Individuals with pre-existing cardiovascular diseases, high blood pressure, diabetes, and metabolic syndrome face an increased risk of mortality [1]. Furthermore, a significant portion of patients experience persistent symptoms and comorbidities that are associated with organ damage. The long-term consequences of these complications remain uncertain.

Understanding the underlying reasons for the significant variation in individual responses to SARS-CoV-2 infection is essential for the development of effective therapies [4]. However, this aspect remains poorly comprehended, posing a major challenge in therapeutic advancements. Recent investigations into the interactions between host cell membranes and the virus have provided valuable insights into the role of cellular lipids in the viral entry process [5]. Specifically, sphingolipids (SLs) have been identified as influential factors in the entry of bacteria and other viruses into cells [6,7]. SLs are essential components of eukaryotic cell membranes, serving as structural elements, signaling molecules, and modulators of enzyme activity [8]. Apart from their structural functions, certain SLs exhibit bioactivity and are associated with various pathological conditions, including inflammation-related disorders like atherosclerosis, rheumatoid arthritis, inflammatory bowel disease, type II diabetes, obesity, cancer, and neurological diseases [9,10,11]. Furthermore, SLs play a critical role in regulating viral replication and the innate immune response [6,12,13]. Due to their versatility, these molecules have extensive involvement in both normal physiological processes and pathological states.

The metabolic processes involving SLs are highly complex. Ceramide (Cer) holds significant importance in the SL metabolic pathway and can be generated through the breakdown of sphingomyelin (SM) or through de novo synthesis [14]. The initial step involves serine palmitoyltransferase, which converts serine and palmitoyl-coenzymeA into 3-ketodihydrosphingosine. Subsequently, 3-ketodihydrosphingosine reductase converts it to sphinganine [14]. Ceramide synthase then adds an acyl-fatty acid to sphinganine, forming dihydroceramide (dH-Cer). Dihydroceramide D4-saturase, located in the endoplasmic reticulum (ER), converts dH-Cer into Cer [14]. Following its production, Cer needs to be transported from the ER to the Golgi organelle to facilitate the synthesis of sphingomyelin (SM) [15]. Moreover, Cer can undergo transformation into various crucial SLs [15]. Ceramidase is an enzyme responsible for converting Cer into sphingosine (Sph) (2-amino-4-trans-octadecene-1,3-diol). Sphingosine kinases 1 (SphK1) and 2 (SphK2) can then phosphorylate Sph to generate the active lipid sphingosine-1-phosphate (S1P) [16]. Sphingomyelin synthase converts Cer into SM, while sphingomyelinase (SMase) converts SM back into Cer. These conversion reactions occur in lysosomes (acid SMase) or on the cell surface (neutral SMase) [8,17].

Emerging evidence suggests that SLs play a crucial role in modulating SARS-CoV-2 infection [5,18]. Studies conducted on animal models have demonstrated that SARS-CoV-2 infection leads to an increase in SL levels, both within cells and in the serum [19]. The significance of SLs extends beyond their role as physical components of cell membranes and ligands. They also have an impact on the localization and activity of proteins involved in receptor-mediated signaling [8]. In patients with COVID-19, plasma concentrations of Cer were found to be elevated [20,21], particularly in those with severe respiratory symptoms [22]. Furthermore, reduced levels of serum Sph were strongly associated with symptomatic COVID-19 compared to asymptomatic cases [23].

In light of previous findings documenting changes in SL patterns among COVID-19 patients, our research takes a step further by integrating targeted mass spectrometry-based sphingolipidomics with the measurement of specific inflammatory mediators, clinical parameters, and the gene expression of SL-metabolizing enzymes. Notably, our study includes patients who have recovered from COVID-19, providing a comprehensive understanding of the disease trajectory. Through this multifaceted approach, our primary objective is to identify biomarkers that can serve as diagnostic and prognostic indicators, while also unraveling the intricate role of SL in the pathophysiological pathways of COVID-19. In line with this, our study emphasizes that SARS-CoV-2 infection induces a notable increase in SM levels in the plasma of patients with severe illness. The implications of these alterations warrant careful consideration, as they could potentially influence disease progression and outcomes.

## 2. Material and Methods

### 2.1. Study Design and Blood Collection

This prospective study took place from June to November 2020, before the COVID-19 vaccination was introduced. We implemented strict and reasonable inclusion and exclusion criteria: adult participants who tested positive for COVID-19 (*n* = 204) through the analysis of nasopharyngeal swabs using a genomic RNA assay with RT-PCR (Biomol OneStep Kit/COVID-19–Institute of Molecular Biology of Parana-IBMP Curitiba/PR, Brazil) or serology-specific IgM and IgG antibody tests (SARS-CoV-2 antibody test^®^, GuangzhouWondfo Biotech Co., Ltd., Guangzhou, China), and control subjects (healthy volunteers–*n* = 55) who tested negative for SARS-CoV-2. Children under 18 years of age and pregnant or lactating women were excluded. Blood samples were collected from patients classified as asymptomatic to mild (*n* = 36), moderate (*n* = 60), severe (*n* = 67), or critical (*n* = 41). The clinical classification criteria were determined at the time of sample collection, based on a modified statement from the Novel Coronavirus Pneumonia Diagnosis and Treatment Guideline (7th edition) [24,25]. At two medical centers, namely Santa Casa de Misericórdia de Ribeirão Preto and Hospital São Paulo in Ribeirão Preto, São Paulo, Brazil, peripheral blood samples were obtained through venous puncture upon patients’ first admission and/or during hospitalization. The blood samples of healthy controls and non-hospitalized participants were taken either at the Centre for Scientific and Technological Development “Supera Park” (Ribeirão Preto, São Paulo, Brazil) or at the residences of patients receiving at-home care. Convalescent participants (*n* = 77) were recruited based on the following inclusion criteria: men and women aged 30 to 69 years, approximately 30 days after the resolution of acute clinical signs of COVID-19 or medical discharge (in the case of hospitalization). The exclusion criteria included acute or chronic clinical illnesses without medical supervision, anemia, use of immunosuppressive drugs, pregnancy, hormone replacement therapy, smoking, and heavy alcohol or drug use. Furthermore, a health status evaluation was conducted. Blood samples were obtained through venipuncture in EDTA tubes using a vacuum collection method (BD Vacutainer^®^ EDTA K2, Franklin Lakes, NJ, USA) from all patients with COVID-19, healthy controls, and convalescent individuals. After centrifugation, plasma and buffy coat (the middle layer containing leukocytes used for genomic RNA extraction) were separated and immediately frozen at −80 °C, either alone or with 0.5 mL of TRIzol reagent, respectively. All blood samples were processed within four hours of collection. For sphingolipidomics analysis, plasma was promptly preserved in methanol (1:1 *v*/*v*) and analyzed using tandem-mass spectrometry (LC-MS/MS).

### 2.2. Ethical Considerations

In compliance with international ethical guidelines, informed consent was obtained from all participants as approved by the National Council of Ethics in Research (CONEP), the Human Ethics Committees of the Faculty of Pharmaceutical Sciences of Ribeirão Preto, and the School of Physical Education and Sport of Ribeirão Preto–University of São Paulo (USP). The research protocol for the Immunocovid study (CAAE: 30525920.7.0000.5403) and AEROBICOVID study (CAAE: 33783620.6.0000.5659 and CAAE: 33783620.6.3001.5403) received the certificate of Presentation and Ethical Appreciation. The sample size was determined based on convenience sampling, the availability of participants at partner hospitals, their willingness to participate, and the local conditions during the pandemic.

### 2.3. Laboratory and Data Collection

Each patient’s electronic medical records were checked thoroughly. The data collected for this study encompassed socio-demographic information, comorbidities, medical history, clinical symptoms, regular laboratory tests, immunological tests, clinical interventions, and outcomes. All information was recorded on a standardized form. The primary endpoint for data collection was the laboratory exams performed within 24 h of admission. The secondary endpoint focused on the clinical outcome, whether the patients were discharged or deceased. For hospitalized individuals, blood exams were carried out in the clinical analysis laboratories of their respective hospitals. As for healthy participants and non-hospitalized patients, blood analyses were conducted at the Clinical Analysis Service (SAC), Faculdade de Ciências Farmacêuticas de Ribeirão Preto, Universidade de São Paulo, Brazil. Automated assays were utilized to assess liver and kidney function, myocardial enzyme spectrum, coagulation factors, red blood cells, hemoglobin, platelets, and total and differential leukocytes.

### 2.4. Cytokine Measurements

Plasma concentrations of interleukin (IL)-6, IL-8, IL-1β, and IL-10 and tumor necrosis factor (TNF) were assessed using a BD Cytometric Bead Array (CBA) Human Inflammatory Kit from BD Biosciences, San Jose, CA, USA, following the manufacturer’s instructions. Briefly, the cytokine beads were counted using a flow cytometer (FACS Canto TM II; BD Biosciences, San Diego, CA, USA), and data analysis was conducted using FCAP Array (3.0) software (BD Biosciences, San Jose, CA, USA). The concentrations of cytokines were reported in pg/mL.

### 2.5. Lipid Extraction and Sample Preparation for LC-MS/MS

All steps were executed at room temperature if not specified otherwise [26,27]. Briefly, plasma samples (250 μL) were regulated to 1 mL with PBS and relocated into a glass centrifuge vial. Then, 10 μL of the internal standard (10 μM Cer/Sph Mixture II, LM6005, Avanti^®^ Polar Lipids, Alabaster, AL, USA), 300 μL of 18.5% HCl, 1 mL of MeOH, and 2 mL of CHCl_3_ were added before the contents were vortexed for 30 min (50 rpm). Samples were centrifuged for 3 min at 2000× *g*, the lower organic phase was moved into a new glass tube, and to the leftover aqueous phase was added another 2 mL of CHCl_3_ to repeat the lipid extraction before combining the two organic phases and vacuum-drying the solvent in a speed-vacuum for 45 min at 60 °C. The sample was resuspended in 100 μL of MeOH:CHCl_3_ (4:1, *v*/*v*) and vortexed for 1 min before being stored at −80 °C.

### 2.6. Sphingolipid Quantification by LC-MS/MS

Plasma samples were analyzed for SL measurements by experimenters who were unaware of the experimental conditions, following previously described methods [26,28]. In summary, liquid chromatography was conducted using an Ascentis Express C18 column (Supelco, St. Louis, MO, USA) with dimensions of 100 × 2.1 mm and a particle size of 2.7 μm maintained at 40 °C throughout the procedure in an ultra-high-performance liquid chromatography (UHPLC) system (Nexera X2; Shimadzu, Kyoto, Japan). The column was equilibrated for 20 min, and then a 10 μL sample was injected onto the HPLC column. Elution was performed using a binary gradient system consisting of Phase A (H_2_O with 1% formic acid) and Phase B (MeOH). The gradient elution lasted 20 min at a flow rate of 0.5 mL/min, with the following settings: 0.0–1 min–30% B, 1.1–2.5 min–85% B, 2.5–5.0 min–100% B, 5.0–15 min–hold 100% B, and 15.1–20 min–re-equilibrated with 30% B. The HPLC system was connected to a TripleTOF 5600+ mass spectrometer (SCIEX, Redwood City, CA, USA). High-resolution multiple-reaction monitoring (MRM^HR^) scanning was performed using an electrospray ionization source (ESI) in positive ion mode. External calibrations of the calibrated delivery system (CDS) were carried out using an atmospheric pressure chemical ionization probe (APCI). Automatic mass calibration (<2 ppm) was performed after each of the five sample injections using APCI Positive Calibration Solution (SCIEX, Redwood City, CA, USA) injected via direct infusion at a flow rate of 300 μL/min. Instrumental parameters included the following: nebulizer gas (GS1) at 50 psi, turbo gas (GS2) at 50 psi, curtain gas (CUR) at 25 psi, electrospray voltage (ISVF) at +4500 V, and turbo ion spray source temperature at 500 °C. The dwell time was set at 10 ms, and a mass resolution of 35,000 was achieved at *m*/*z* 400. Data acquisition was performed using Analyst^TM^ software (SCIEX, Redwood City, CA, USA). Qualitative identification of lipid species was carried out using PeakView^TM^ software (SCIEX, Redwood City, CA, USA), and the mass transitions for all standards and analytes were reported in Appendix A, including typical retention times. For quantitative analysis, MultiQuant^TM^ software (SCIEX, Redwood City, CA, USA) was used, enabling the normalization of the peak area of individual molecular ions using an internal standard for each family of SLs (Cer/Sph Mixture II-Avanti Polar Lipids-LM6005). The raw area of the analytes was monoisotopically corrected (Appendix A) to account for the natural abundance of ^13^C and ^14^C isotopes, which were not considered in the analytical method [29]. The quantification of SL species involved calculating the area ratio of each lipid to the corresponding internal standard, multiplied by the concentration of the internal standard (Appendix A), resulting in the actual analyte concentration (pmol/mL) in plasma. If no standards were available for a specific SL species, its quantification was performed using the internal standard concentration of the closest counterpart. Additional details on the methodology can be found in Appendix A. 

### 2.7. RNA Extraction and Analysis

The buffy coat was processed via two stages in order to obtain total RNA. Initially, the samples were defrosted, and then 0.5 mL of TRIzol reagent (Sigma, San Luis, MO, USA) was added followed by a 5 min incubation at room temperature. Next, 0.2 mL of CHCl_3_ was added to the mixture, which was then incubated for 3 min at room temperature. Subsequently, the mixture was centrifuged at 12,000× *g* for 15 min at 4 °C. The upper phase, containing the RNA, was carefully transferred to a new tube, and an equal volume of ethanol (1:1 *v*/*v*) was added. After this initial stage, 0.7 mL of each sample was transferred to a spin cartridge containing a clear silica membrane for further extraction with on-column DNase treatment, following the manufacturer’s specifications (Life Technologies, PureLink RNA Mini Kit, Catalog No. 12183018A, Carlsbad, CA, USA) and using the PureLink DNase Set (Life Technologies, Catalog No. 12185010). The concentration of RNA was determined using a Qubit Fluorometer (Life Technologies, Qubit RNA BR Assay Kit, Catalog No. Q10211). The purity of the RNA was assessed by measuring the absorbance ratios at 260/280 nm and 260/230 nm using a Nanodrop spectrophotometer (Thermo Fisher Scientific, Waltham, MA, USA). The quality of the purified total RNA was evaluated by determining the RNA Integrity Number (RIN) values using a Bioanalyzer instrument (Agilent Technologies, Agilent 2100 Bioanalyzer system with RNA 6000 Nano kit, Catalog No. 5067-1511, Santa Clara, CA, USA).

### 2.8. Transcriptome Profiling

The Clariom S Human Assay (Applied Biosystems, Clariom S Assay human, Catalog No. 902927, Foster City, CA, USA) was used for single-sample (cartridge array) processing on the GeneChip 3000 instrument system (Applied Biosystems, GeneChip WT Pico Reagent Kit, Catalog No. 902622). This process was carried out in a high-throughput facility, specifically the Thermo Fisher Scientific Microarray Research Services Laboratory located in Santa Clara, CA, USA. Using this setup, whole-transcript expression arrays were generated from a total of 66 samples. Among these samples, there were 12 healthy controls and 54 patients diagnosed with COVID-19.

#### 2.8.1. Bioinformatic Analysis of Transcriptome Data

Gene expression data for enzymes involved in the SL pathway were obtained from the ArrayExpress database (http://www.ebi.ac.uk/arrayexpress accessed on 22 June 2022) under accession number E-MTAB-11240. This dataset included preprocessed transcriptomic profiling from a total of 66 samples, consisting of 12 healthy controls and 54 patients diagnosed with COVID-19. The patient group was further subdivided based on clinical classification: mild (*n* = 12), moderate (*n* = 14), severe (*n* = 14), and critical (*n* = 14). Bioinformatic analyses were conducted using R 4.1.2 libraries [30] in the RStudio environment [31] and Bioconductor libraries [32]. The expression data at the probe set level were preprocessed, and gene-based expression was obtained by collapsing the probes using the maximum mean method with the collapseRows function of the WGCNA 1.71 package. The gene-to-probe annotation available in the same dataset was used for this purpose. Differential expression analysis for the whole transcriptome between the different clinical groups mentioned above was performed using the limma 3.50.1 package. Age, sex, body mass index (BMI), hypertension, diabetes, and outcome were included as co-variates in the analysis. Differentially expressed genes (DEGs) were defined based on adjusted *p*-values (Benjamini and Hochberg method) less than 0.05 in at least one pair of clinical groups [33]. Graphical representations of the generated data were constructed using the ggplot2 3.3.5 package.

#### 2.8.2. Validation of Microarray Data by Reverse Transcription Quantitative Real-Time PCR

For confirmation of RNA samples used in microarray-based gene expression profiling, a two-step RT-qPCR (Reverse Transcription quantitative PCR) approach was employed. Firstly, complementary DNA (cDNA) was synthesized from 300 ng of total RNA using the High-Capacity cDNA Reverse Transcription Kit (Applied Biosystems^TM^), following the manufacturer’s instructions. The RT-qPCR reactions were prepared using 1X iTaqTM Universal SYBR^®^ Green Supermix (BIO-RAD), 100 nM of each gene-specific primer combination, and 8 ng of cDNA. These reactions were then amplified in an AriaMX Real-time PCR machine (Agilent Technologies, Santa Clara, CA, USA). The RT-qPCR cycling protocol included an initial denaturation step at 95 °C for 30 s, followed by 40 cycles of denaturation at 95 °C for 5 s and annealing/extension at 60 °C for 20 s. Melting curve analysis was performed to ensure the absence of non-specific reactions. To measure the efficiency of primer pairs, the Real-time PCR Miner algorithm was utilized [34]. Each sample was tested in duplicate, and gene expression analyses were performed using the ΔCq model with β-Actin serving as the reference gene [35].

### 2.9. Statistical Data Analysis

The data obtained from the study were evaluated for normal distribution using the Shapiro–Wilk normality test and D’Agostino and Pearson test. Parametric data were analyzed using unpaired *t*-tests or one-way ANOVA, followed by Tukey’s multiple comparison tests. Non-parametric data were analyzed using Mann–Whitney or Kruskal–Wallis tests, followed by Dunn’s post-tests. The chi-square test was used to assess associations among categorical variables and COVID-19. Statistical analysis was performed using GraphPad Prism software version 9.0, and statistical significance was set at *p* < 0.05. The dependence on multiple variables was analyzed using Spearman’s correlations, with statistical significance set at *p* < 0.05. The correlation matrix was presented using the R package qgraph [36]. Partial least squares discriminant analysis (PLS-DA) was conducted using the MetaboAnalyst 5.0 online software (https://www.metaboanalyst.ca/ accessed on 23 May 2023.). The relevance of sphingolipid metabolites was assessed using the variable importance in projection (VIP) score, and potential biomarkers were identified based on this score [37]. The accuracy of the predictor was determined by calculating the area under the curve (AUC) of the receiver operating characteristic (ROC). The AUC, along with its 95% confidence interval, was used to assess the diagnostic value. An AUC > 0.70 was considered clinically relevant. Multivariate binomial logistic regression analysis was performed using the Jamovi software (Version 1.6-2021) to assess the association of SLs with the prognosis and clinical outcomes, specifically severity and mortality. The regression model was adjusted for confounding variables such as age, sex, comorbidity, body mass index (BMI), hospitalization days, blood glucose, and neutrophil–lymphocyte ratio (NLR). 

## 3. Results

### 3.1. Characterization of Study Participants

Patients’ blood samples were collected during the height of the pandemic in the clinical emergency setting, ensuring a diverse representation without specific selection or matching. A total of 204 patients diagnosed with COVID-19 were included in this study, categorized based on the severity of their symptoms: mild (*n* = 36), moderate (*n* = 60), severe (*n* = 67), and critical (*n* = 41). These groups were compared to convalescent patients (*n* = 77) and a healthy control group (*n* = 55). Detailed demographic characteristics, clinical manifestations, and laboratory findings of the patients, convalescent individuals, and control group are provided in Table 1. As expected, the groups differed significantly in terms of age, with advanced age being a well-established risk factor for hospitalization and severity of COVID-19 infection. There was a significant difference in the distribution of sex among the convalescent group. The most prevalent comorbidities observed in COVID-19 patients were hypertension (44.1%), diabetes (30.1%), cardiovascular disorders (21%), increased body mass index (BMI) (28.4 ± 5.9), history of smoking (19%), and neurological disorders (17%). The common initial symptoms reported by patients included cough, dyspnea, dysgeusia, and anosmia, followed by diarrhea, fever, myalgia, and hyperactive delirium.

Regarding hematological characteristics, severe and critical patients exhibited a significant decrease in hemoglobin, erythrocyte count, and lymphocyte count, along with a significant increase in total leukocyte count, neutrophil count, neutrophil–lymphocyte ratio (NLR), and blood glucose levels compared to patients with mild/moderate symptoms and the control group. Hematological parameters were assessed only at the time of admission. Oxygen saturation was significantly lower in COVID-19 patients compared to convalescent individuals and the control group. Approximately 65% of patients required oxygen administration, with the majority receiving oxygen therapy via a Venturi Mask (17.1%), high flow nasal oxygen support (31.9%), or invasive mechanical ventilation (16.2%). The average hospitalization duration for severe and critical patients was 9 ± 4 days. In terms of medical treatment, glucocorticoids were prescribed to 61% of patients, Azithromycin to 59.3%, Ceftriaxone to 45.6%, Oseltamivir to 29.4%, Hydroxychloroquine to 13.2%, and anticoagulants to 8.8%, aligning with the standard care protocols in place during the patients’ hospitalization period.

### 3.2. COVID-19 Severity Increased Gene Expression of Key Enzymes Involved in SM and Cer Synthesis

The metabolism of SLs is highly intricate [8]. We sought to investigate whether the gene expression of enzymes involved in their production was regulated in blood cells from the buffy coat. Cer serves as a crucial component in the metabolic flux of SLs and can be generated through either de novo synthesis or the breakdown of SM. The de novo pathway begins with the action of serine palmitoyltransferase (SPT). *SPTLC2* showed upregulation in severe and critical patients, while *SPTLC1* and *SPTLC3* were primarily elevated in critical patients (Figure 1A). Sequential reactions introduce variations to the core structure, leading to Cer formation. The enzyme sphingolipid-4-desaturase (*DEGS1*) adds a double bond to the sphingoid base, while *DEGS2* incorporates a hydroxide group (OH). Notably, *DEGS1* expression was high in severe cases (Figure 1B), and *DEGS2* displayed a tendency to decrease with severity (Appendix A). In the catabolic pathways of SLs, SM, ceramide-1-phosphate (C1P), and glycosphingolipids are hydrolyzed, resulting in Cer formation. The expression levels of Cer-metabolizing enzymes derived from SM, such as sphingomyelinases (*SMPD2* and *SMPD3*), were found to be downregulated in COVID-19 severity (Figure 1C). Furthermore, the expression levels of other enzymes associated with this pathway, such as ceramide synthase 2 (*CERS2*) and ceramide synthase 4 (*CERS4*) (Figure 1E), as well as the ceramide transfer protein (*CERT1*) (Figure 1F), were increased in severe COVID-19 patients. However, no significant regulation was observed in the gene expression of *CERS1*, *CERS3*, *CERS5*, and *CERS6* within the same COVID-19 group (Appendix A). Cer can be converted into several important SLs. The enzyme ceramidase generates sphingosine (Sph), and the expression of acid ceramidase 1 (*ASAH1*) was elevated in critical patients (Figure 1D), while *ASAH2* showed no influence (Appendix A). Conversely, the expression levels of enzymes involved in the conversion of Cer to SM, such as sphingomyelin synthase 1 (*SGMS1*), were significantly increased in COVID-19 patients in a severity-dependent manner (Figure 1C). Intriguingly, *SGMS2* demonstrated an increase only in critically ill patients (Figure 1C). Moreover, the majority of genes that promote Cer accumulation exhibited slight expression levels in severely affected patients, whereas genes involved in Cer-to-SM synthesis displayed significant upregulation.

### 3.3. Plasma SM Profile Is Associated with COVID-19 and Can Be a Potential Biomarker for Assessing Severity of Disease

The plasma SL profile of the study participants was determined using LC-MS/MS targeted sphingolipidomics (Figure 2). A total of thirty-eight SL species, classified into seven subclasses, were identified in the plasma of the volunteers. Appendix A provide a comprehensive list of the measured lipids, including their annotations, retention time, and fragment panel. The data points were organized into distinct groups for analysis: (i) COVID-19 patients categorized according to disease severity (ranging from asymptomatic to mild, moderate, severe, and critical); (ii) healthy control subjects; and (iii) convalescent individuals (30 days post COVID-19 recovery). Examining the overall SL plasma composition in the control group, SM was found to be the predominant species (91%), followed by Cer (7%), dihydroceramide (dHCer) (0.4%), lactosylceramide (LaCer) (0.2%), sphingosine (Sph) (0.06%), hexaglycosylceramide (HexCer) (0.05%), and sphinganine (0.01%) (Figure 2A). Comparing the SL profiles of COVID-19 patients with those of the control group, we observed an increased percentage of SM production correlated with the severity of symptoms. The highest production was observed in the plasma of severe patients (96.64%), accompanied by slightly decreased total levels of other SL classes, specifically Cer, which exhibited reduced percentage levels in patients with more severe clinical manifestations. Interestingly, the total SL percentage profile of convalescent individuals closely resembled that of the control group, indicating a restoration of lipid homeostasis (Figure 2A).

In quantifying the production of each SL class in plasma (measured in pmol/mL) we found a statistically significant increase in the total production of SM species in COVID-19 patients compared to both the control and convalescent groups (Figure 2C). Moreover, we demonstrated that statistical differences existed between the different severity groups of COVID-19 patients, control subjects, and convalescent individuals for sphinganine (Appendix A), dHCer (Appendix A), Cer (Appendix A), LacCer (Appendix A), HexCer (Appendix A), and Sph (Appendix A). These differences underscored the increased production of Cer and its derivative metabolites (dHCer, LacCer, and HexCer) in severe and critical COVID-19 patients. In this sense, several SL classes increased their production with disease severity, but SM levels seem to be more prominent in the total percentage ratio. The analysis of the principal metabolic species associated with sphingolipid (SL) subclasses revealed the predominant identification of 38 lipids, represented in the heatmap plots based on COVID-19 severity (Figure 2B). The average values of each SL species across all groups showcased a significant remodeling of plasma SL species in individuals with COVID-19, with certain species exhibiting prevalence by specific groups. 

Given the extensive sphingolipidomic data, we conducted a comprehensive analysis to identify potential biomarkers and a signature associated with the progression of COVID-19. Initially, we focused on discerning the differences between severe COVID-19 patients and healthy participants. The 3D score plot in Figure 2D, generated through partial least squares discriminant analysis (PLS-DA), clearly demonstrates the presence of a distinct sphingolipid (SL) profile associated with the disease. To identify the most predictive and differentiating features that could aid in sample classification, we utilized the variable importance in projection (VIP) score, which is derived from PLS-DA and signifies the key factors contributing to group disparities (Figure 2E). A VIP score greater than 1 indicates that a metabolite may serve as a potential biomarker [38]. Notably, the subclass of SLs known as total SM exhibited the highest VIP score, suggesting its potential as a biomarker for assessing the severity of COVID-19 (Figure 2E). Following the identification of prospective biomarkers using the VIP score plots, we further evaluated their effectiveness and predictive power in distinguishing severe COVID-19 groups from controls using a receiver operating characteristic (ROC) curve analysis. In this regard, we specifically focused on total SM (Figure 2F). The area under the curve (AUC) for total SM exceeded the threshold considered clinically significant (AUC = 0.81, *p* < 0.0001). These findings highlight the potential utility of SM as a biomarker for assessing the severity of COVID-19.

### 3.4. The Discovered Plasma SM Species Panel Effectively Distinguished Severe COVID-19 

After conducting PLS-DA and VIP score analyses for total SM, we proceeded to assess the significance of individual SM species using the same statistical method (Figure 3A,B). Remarkably, we identified SM (d18:1/24:0), (23:0), (16:0), (24:1), (25:1), (23:1), (25:0), (26:0), and (26:1) with VIP scores exceeding 1.4 in the severe patient group, indicating their potential as biomarkers for COVID-19 severity (Figure 3B). To further investigate these altered SM compounds, we employed ROC curves (Figure 3C–L) and boxplot graphics (Figure 3M–V). All ten selected SM species exhibited excellent diagnostic performance (AUC > 0.70; *p* < 0.0001), effectively distinguishing severe COVID-19 patients from healthy controls (Figure 3C–L). Additionally, the median levels of these SM biomarkers exhibited significant increases across control, mild, moderate, severe, and critical COVID-19 patient groups (Figure 3M–V). Notably, in plasma samples from convalescent individuals, the levels of SM (d18:1/16:0), (24:1), (25:1), (26:0), and (26:1) showed a substantial elevation compared to severe and critical patients (Figure 3M,R,T,U,V). Other SM species with moderate or low VIP scores (<1.4) were evaluated for biomarker potential using the ROC curve analysis (Appendix A). SM (d18:1)/16:1, 18:0, 20:0, 20:1, and 22:0 demonstrated diagnostic suitability (AUC > 0.70; *p* < 0.0001), while SM (d18:1)/14:0 did not (Appendix A). Importantly, the production of these SM species increased with the severity of COVID-19 and subsequently decreased in convalescent individuals (Appendix A), thereby supporting the association between SM and disease severity. Further, we assessed whether the production of total SM species, as well as SM (24:0) specifically, is affected by glucocorticoid use and gender. Among the patients, 61.3% received glucocorticoid therapy as part of their treatment. The analysis revealed that the production of total SM species and SM (24:0) was not significantly affected by glucocorticoid use (Appendix A) or gender (Appendix A). This finding supports an association concerning SM levels and disease severity in addition to other patient confounding variables. Nevertheless, further confirmation of these findings requires multivariate analysis to determine their impact on disease outcome.

### 3.5. Multivariate Binomial Logistic Regression Determines the Association between Cer/SM Species and COVID-19 Clinical Severity and Mortality 

The SM species previously identified as potential biomarkers for COVID-19 were further examined to determine their relationship with disease severity (Figure 4A) and mortality outcomes (Figure 4B) using multivariate binomial logistic regression. The statistical model was adjusted for established risk factors including age, sex, comorbidities, blood glucose, and NLR (neutrophil–lymphocyte ratio). Similarly, the same statistical model was applied to assess the association of Cer species, which exhibited high production in the plasma of COVID-19 patients. Our findings indicated that the species Cer (d18:1/24:1) and Cer (d18:1/24:0) remained important predictors of symptom intensity, with respective odds ratios of [3.82 (CI: 0.15–99), *p* = 0.422] and [0.39 (CI: 0.01–8.20), *p* = 0.552]. The regression model, which demonstrated statistical significance (*p* < 0.001) and yielded high values for R^2^MacFadden (0.8) and R^2^Nagelkerke (0.9), along with a specificity of 0.93, sensitivity of 0.89, and an AUC of 0.98, suggested a possible protective effect of SM (d18:1/24:0) against disease severity [0.58 (CI: 0.36–0.92), *p* = 0.022] (Figure 4A). Conversely, the levels of SM (d18:1/24:0) did not show a significant association with patient discharge from the hospital (Figure 4B). The significant regression model (*p* < 0.001), with R^2^MacFadden = 0.6, R^2^Nagelkerke = 0.5, specificity = 0.91, sensitivity = 0.64, and an AUC of 0.92, indicated that the Cer species (d18:1/24:1) and (d18:1/24:0) may be linked to mortality, with odds ratios of [1.21 (CI: 0.55–2.63), *p* = 0.635] and [0.51 (CI: 0.07–3.69), *p* = 0.505] (Figure 4B). However, no other SM species showed a substantial association with disease severity or death in COVID-19, as determined by the regression analyses.

### 3.6. Correlation of Values of SM Species with Immunological, Clinical, and Laboratory Markers in COVID-19 

To investigate the relationship between plasma levels of SM species and clinical and laboratory features, as well as inflammatory mediator production, we employed Spearman’s test. This analysis aimed to unravel the interplay between these lipid compounds and other parameters associated with the immunopathogenesis of COVID-19 (Figure 4C). Following the identification and validation of SM (d18:1/16:0), (24:1), and (24:0) and Cer (d18:1/24:1) and (24:0) as potential significant metabolites in relation to COVID-19, we proceeded to examine their associations. Our results revealed that both SM species and Cer species exhibited positive correlations with age, number of hospitalization days, clinical scores, glycemia, neutrophil count, NLR, and the production of IL-10, IL-6, and IL-8. Conversely, they demonstrated negative correlations with oxygen saturation at the time of admission, hemacytometer count, and lymphocyte count. In terms of the strength of these associations, SM species displayed stronger and statistically significant connections to inflammatory parameters compared to Cer species. This finding suggests their involvement in the immunopathological processes of COVID-19. However, it is important to note that SM species may not necessarily contribute to disease severity. Instead, they might serve as counterpoint factors, influencing the balance of inflammation during the development of COVID-19. All the Spearman’s test statistical values are shown in Appendix A.

## 4. Discussion

Given the intricate connection between lipids and the processes involved in the development of infectious diseases, it is plausible to explore the potential of analyzing alterations in the plasma lipid profile as a means to identify biomarkers. In this study, we employed our quantitative sphingolipidomics technology to evaluate clinical and inflammatory markers associated with plasma levels of SLs and the expression of enzymes involved in the SL pathway in blood cells. Our sample consisted of a larger group of COVID-19 patients with varying degrees of severity, carefully characterized, and compared to convalescent individuals. While previous research on COVID-19 has predominantly focused on Cer, with only a limited number of studies investigating other SL metabolites, our findings present a novel discovery. We observed a progressive increase in the levels of SM class metabolites with the severity of the disease. Specifically, we identified elevated plasma levels of long-chain SM (d18:1) in severe COVID-19 patients when compared to those with mild symptoms and convalescent individuals. Interestingly, severe patients exhibited a higher proportion of SM in their overall SL output compared to patients with moderate symptoms and critical cases. This observation may indicate a pivotal point in the immunopathogenesis pathway of COVID-19, suggesting a bias towards SM synthesis in Cer metabolism.

To gain insight into the underlying mechanisms of SL production during COVID-19, we analyzed gene expression patterns of a comprehensive panel of enzymes involved in the SL pathway in patients’ peripheral blood cells, as the biochemical activity assays for these enzymes are yet to be developed. Overall, we identified altered expression of enzymes associated with de novo SL synthesis. Notably, serine palmitoyltransferases (*SPTLC1*, *SPTLC2*, and *SPTLC3*) were upregulated in severe and critical forms of the disease, potentially contributing to the increased production of sphinganine and dHCer. Additionally, the expression of desaturase (*DEGS1*) was enhanced, leading to enhanced Cer production [11]. The Cer molecules could then undergo modifications, such as the addition of phosphocholine by sphingomyelinase pathway enzymes like SM synthase (*SGMS1* and *SGMS2*) [38], which catalyze the conversion of Cer to SM. These enzymes were also upregulated in severe and critical COVID-19 patients. Alternatively, ceramidases (*ASAH1*), upregulated in critical illness, could deacylate Cer to form Sph [39]. Notably, in severe forms of COVID-19, ceramide synthases (*CERS2*, *CERS4*, and *CERT1*) responsible for the synthesis of Cer from Sph [40] were upregulated. However, the expression of sphingomyelinases (*SMPD2* and *SMPD3*), which catalyze the formation of Cer from SM [5], tended to decrease with increasing disease severity. Our data suggested an increased flux of metabolites towards Cer synthesis through multiple pathways, followed by elevated SM synthesis originating from this central pool of Cer, correlated with COVID-19 development. 

SM is the primary SL present in cell membranes. Acid sphingomyelinase activity (aSMase), an enzyme found in lysosomes and in the cell membrane, is responsible for the conversion of SM to Cer. The aSMase/Cer system actively participates in host defense responses [7,41,42]. SM, being a ubiquitous component of cells, is involved in various cellular activities, including cell division, proliferation, and autophagy [43]. It also helps maintain a balance between pro-inflammatory and anti-inflammatory lipids, thus regulating the immune system in lung tissues [44]. On the other hand, Cer predominantly participates in processes related to inflammation and damage [45]. Cholesterol, SM, and phosphatidylcholines (PC) have implications for the immune system and play critical roles in macrophage activation [46], NK cell function [47], and the development and activity of T and B effector cells [48]. Cer species, specifically, are recognized as pro-inflammatory lipids in lung epithelial cells. In sepsis mouse models, inhibiting Cer synthesis by targeting aSMase through pharmacological means has shown potential in reducing organ damage caused by reactive oxygen species and inflammation [49,50]. Additionally, Cer is involved in insulin signaling [51] and metabolic disorders like obesity in humans [52]. This observation is intriguing since comorbidities such as obesity and type 2 diabetes are associated with worse outcomes in COVID-19. In this context, we observed a tendency to privilege SM synthesis over Cer in severe patients, but this event was not similarly observed in critical patients, suggesting that SM could be involved in a putative host mechanism to recover the disease development from a worst prognostic.

Our findings also indicate a decrease in total SM counterparts and an increase in sphinganine, Sph, and Cer species in convalescent patients, consistent with reports of elevated levels of these SL species in symptomatic individuals compared to asymptomatic ones [23]. This suggests a potential role involved in inflammation control. Moreover, we conducted ROC and correlation analyses to assess the relationship between these SL classes and the severity of COVID-19. We discovered that total plasma levels of SM with an AUC greater than 0.80 (*p* < 0.001) have the potential to identify clinically severe symptoms. In fact, our findings demonstrated that SM (d18:1/16:0), (22:1), (23:0), (23:1), (24:0), (24:1), (25:0), (25:1), (26:0), and (26:1) may serve as predictors of a poor prognosis (AUC > 0.7) for COVID-19. Additionally, the results demonstrated a moderate positive correlation between total SM and inflammation markers in COVID-19, with a high disease clinical score. It is crucial to identify the mechanism by which SARS-CoV-2 increases SL levels. SM has the ability to bind with cholesterol and form lipid rafts, potentially facilitating virus entry into cellular surfaces [53,54,55]. Conversely, the depletion of host and viral SM has been observed to inhibit influenza virus infection [56]. On the other hand, previous studies [20,22,23,57] have reported elevated levels of Cer and dHCer during COVID-19. 

We have identified novel potential SM biomarkers that could predict the progression of COVID-19 towards severe symptoms. Specifically, the SM species (d18:1/24:0) and (d18:1/24:1) were evaluated as potential severity biomarkers in our study. To account for confounding variables such as age, sex, comorbidities, body mass index (BMI), hospitalization days, glycemia, and the neutrophil–lymphocyte ratio (NLR), we used multivariate binomial logistic regression to assess their association with disease severity. Interestingly, our analysis revealed a tendency toward a protective effect mediated by SM (24:0) against severe forms of COVID-19, as well as a risk factor mediated by Cer (24:1), based on a significant regression model (AUC = 0.98; *p* < 0.001). In fact, elevated levels of SM (24:1) were positively correlated with clinical and inflammatory indicators in COVID-19, including the clinical score, hospitalization days, and neutrophil count, as well as the production of IL-10, IL-6, and IL-8. On the other hand, the correlation between Cer (24:1) and these indicators was weaker. These systemic SM species hold promise as potential indicators or targets for preventing disease progression and exploring new treatments. They may have direct effects on regulating inflammation, coagulopathy, and the cytokine storm, which are defining characteristics of severe COVID-19 [58,59]. Furthermore, the combination of SL species production and expression of enzymes involved in SL metabolism shows diagnostic potential and can assist in prioritizing individuals for therapy with newly developed antiviral medications [60,61].

Our study distinguishes itself from previous research by including a larger and more extensively characterized cohort of COVID-19 patients, encompassing healthy controls and individuals in the convalescent phase. However, it is crucial to acknowledge certain limitations that should be addressed in future investigations. Firstly, we have not accounted for the impact of chronic disorders and secondary infections, which could potentially contribute to the dysregulation of SL metabolism. These factors may influence the observed SL patterns and should be considered in future studies. Additionally, it is important to note that our sample is regionally limited, which may introduce geographic and demographic biases. To establish the predictive effect of SL biomarkers in critically ill individuals with COVID-19, it is necessary to conduct cohort studies with broader representation and diverse populations. Furthermore, to gain a comprehensive understanding of SL dynamics, it would be advantageous to assess the SL profile over the course of the disease. This longitudinal analysis would allow us to examine how SL patterns evolve and whether early alterations can serve as predictive markers for disease progression and outcome. Such investigations could provide valuable insights into the pathogenesis of COVID-19 and aid in the development of targeted interventions. By considering additional factors, expanding the scope of the study population, and conducting longitudinal analyses, we can enhance our understanding of the predictive potential of SL patterns and their implications for patient management and therapeutic interventions. 

## Figures and Tables

**Figure 1 cells-12-01938-f001:**
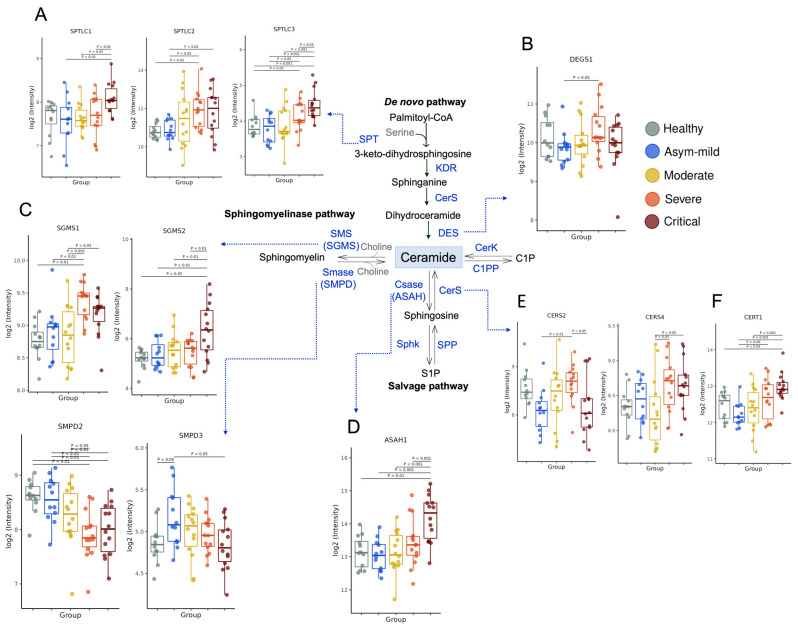
Altered genetic pattern of expression of sphingolipid metabolic enzymes in patients with COVID-19 according to severity. Schematic representation of sphingolipid formation pathways: (**A**) serine palmitoyltransferase, (*SPTLC1/2/3*; (**B**) sphingolipid delta (4)-desaturase DES1 (*DEGS1*); (**C**) sphingomyelin synthase (*SGMS1/2*) and sphingomyelin phosphodiesterase (SMPD2/3); (**D**) N-acylsphingosine amidohydrolase 1 (*ASAH1*); (**E**) ceramide synthase (*CERS2/4*); (**F**) ceramide transporter 1 (*CERT1*) in control (*n* = 12), mild (*n* = 12), moderate (*n* = 14), severe (*n* = 14) and critical (*n* = 14). SM: sphingomyelin; SPT: serine palmitoyltransferase; KDR: 3-keto-dihydrosphingosine reductase; CerS: ceramide synthase; CSase: ceramidase; Des: desaturase; SphK: sphingosine kinase; SPP: S1P phosphatase; SMase: sphingomyelinase; SMS: sphingomyelin synthase; CerK: ceramide kinase; C1PP: ceramide-1-phosphate phosphatase. The log_2_ of normalized gene expression profiles for analyzed groups are showed as boxplots. Significant differences in transcript expression correspond to Benjamini and Hochberg adjusted *p*-values obtained from whole transcriptome differential expression analysis considering a threshold of <0.05 in at least one pair of clinical groups. Details of enzyme nomenclatures Appendix A.

**Figure 2 cells-12-01938-f002:**
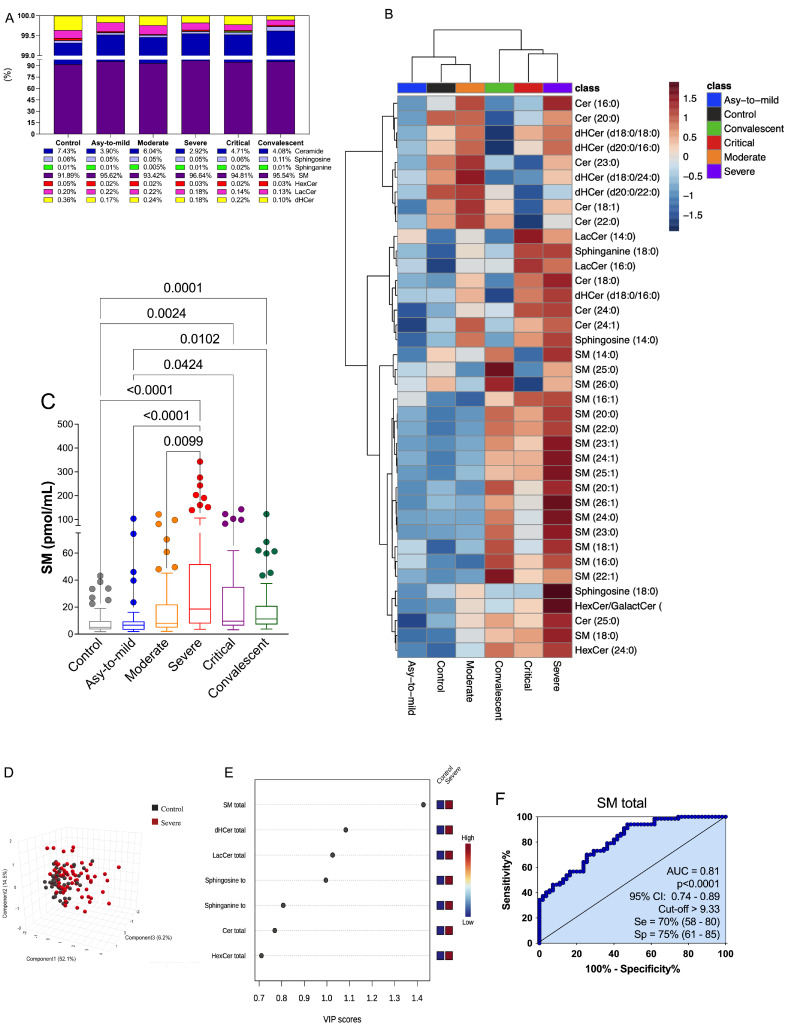
Dynamics of sphingolipid abundance reveal changes in bioactive sphingolipid metabolism and clinical prognosis in patients with COVID-19. (**A**) Relative abundance of sphingolipid metabolite in controls (*n* = 55) and according to the severity of COVID-19 in mild (*n* = 36), moderate (*n* = 60), severe (*n* = 67), and critical (*n* = 41) patients and convalescent individuals (*n* = 77). (**B**) Hierarchical clustering result shown as heatmap based on resource intensity of metabolic species related to sphingolipid subclasses in COVID-19 severity. (**C**) LC–MS/MS measurements for sphingomyelin (SM) in control subjects compared to patients with COVID-19 and convalescents. (**D**) Representation of the three-dimensional dispersion of the main components of the data matrix by a 3D score plot in the comparative groups control (*n* = 55) and severe (*n* = 67) in relation to sphingomyelin. (**E**) Screening by analyzing the VIP score graph (VIP: variable importance in the projection) based on the order of the variables in component 1 and relating the relevance of each variable (VIP cutoff > 1). (**F**) ROC curves for total sphingomyelin in patients with severe forms of COVID-19. The curves compare the severe patient group (*n* = 67) with the control (*n* = 55). AUC: area under the curve; Se: sensitivity; Sp: specificity; CI: 95% confidence interval. (**C**) Statistical analyses were performed using the Kruskal–Wallis multiple comparison test (non-parametric) followed by the Dunn post-test for pairwise comparison. Data are expressed as median in boxplot graphics. Significance levels shown are based on statistically significant between groups with *p*-values < 0.05.

**Figure 3 cells-12-01938-f003:**
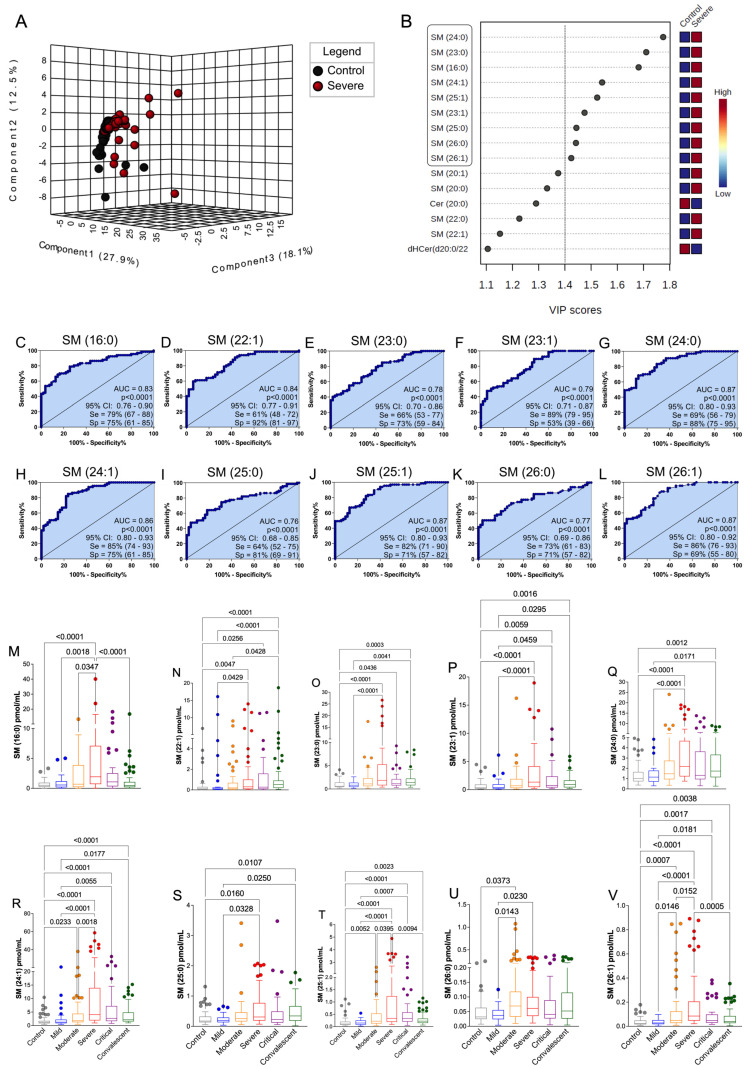
Analysis of serum abundance of sphingomyelin (SM) species as biomarkers of severity in COVID-19. (**A**) Three-dimensional score plot represented as a three-dimensional scatter plot of the 2 main components in the data matrix (control *n* = 55 and severe *n* = 67). (**B**) Projection graph (VIP score) with the order of the variables of component 1 important in the classification of potential biomarkers responsible for the variation between groups (VIP > 1.4). (**C**–**L**) ROC curves for sphingomyelin species (SM) in patients with severe forms of COVID-19. The curves compare the severe patient groups (*n* = 67) with the controls (*n* = 55). AUC: area under the curve; Se: sensitivity; Sp: specificity; CI: 95% confidence interval. (**M**–**V**) Sphingomyelin class profile in control subjects compared to patients with COVID-19 and convalescents in mild (*n* = 36), moderate (*n* = 60), severe (*n* = 67), critical (*n* = 41), and convalescent individuals (*n* = 77). Statistical analyses were performed using the Kruskal–Wallis multiple comparison test (non-parametric), followed by the Dunn post-test for pairwise comparison. Data are expressed as median in boxplot graphics. Significance levels shown are based on statistically significant values between groups with *p*-values < 0.05.

**Figure 4 cells-12-01938-f004:**
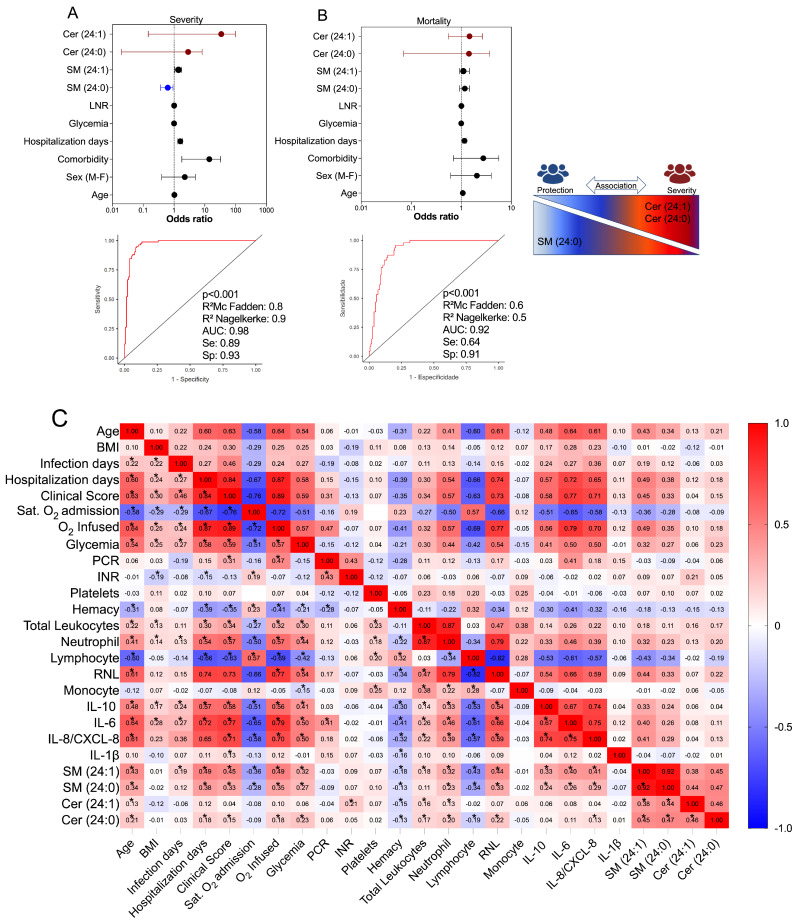
Association of sphingomyelin (SM) and ceramide (Cer) species with severity and mortality outcomes in patients with COVID-19. (**A**) Multivariate analysis of binomial logistic regression for severity and (**B**) mortality outcome, adjusting the model for age, sex, comorbidity, hospitalization days, blood glucose, NLR, and ROC curve evaluating the discrimination capacity of the presented regression models (control *n* = 55, mild *n* = 36, moderate *n* = 60, severe *n* = 67, and critical *n* = 41). NLR: neutrophil–lymphocyte ratio. OD: odds ratio. CI: Confidence interval. (**C**) Correlation matrix demonstrating interactions between SM (24:0), SM (24:1), Cer (24:0), and Cer (24:1) and inflammatory and clinical parameters. Color scale sidebar indicates correlation coefficients (r) color coded as follows: red, positive correlation; blue, negative correlation. The intensity of the color represents the intensity of the correlation. Values vary between −1.0 and 1.0. The significance levels indicated with gray asterisks are based on the *p*-value < 0.05 of the Spearman’s correlation coefficient (R) * and detailed in Appendix A.

**Table 1 cells-12-01938-t001:** Clinical and laboratory assessments of study participants.

Variables	Healthy Controls *n* = 55	COVID-19 Patients *n* = 204	COVID-19 Patients					*p*-Value
			Asy-to-Mild *n* = 36	Moderate *n* = 60	Severe *n* = 67	Critical *n* = 41	Convalescent *n* = 77	
**Demographic characteristics**								
Age, M ± SD, and (IQR)	35 ± 12.9 (19–69)	55 ± 19 (20–96)	37.5 ± 11.4 (21–67)	49 ± 18 (24–92)	63 ± 15.9 (30–96)	71 ± 17.1 (20–94)	46 ± 9.7 (30–66)	^a,d,e^ < 0.0001; ^c^ 0.0002 ^f^ 0.0041
Age < 50, n (%)	45 (81.8)	79 (38.7)	28 (77.8)	31 (51.7)	17 (19.4)	7 (17.1)	42 (54.5)	^a,d,e^ < 0.0001, ^c^ 0.0008 ^f^ 0.0014
Age ≥ 50, n (%)	10 (18.2)	125 (61.3)	8 (22.2)	29 (48.3)	54 (80.6)	34 (82.9)	35 (45.4)	
**Sex, n (%)**								
Man	24 (43.6)	116 (56.9)	15 (41.7)	36 (60)	40 (59.7)	25 (61)	9 (11.7)	^f^ < 0.0001
Woman	31 (56.4)	88 (43.1)	21 (58.3)	24 (40)	27 (40.3)	16 (39)	68 (88.3)	
BMI (kg/m^3^)	25.4 ± 4.2 (15.4–34.9)	28.4 ± 5.9 (15.8–50.3)	27.8 ± 5.3 (15.8–43.8)	28.3 ± 5.7 (17.4–42.1)	28.1 ± 6.1 (20.2–47.7)	29.4 ± 6.1 (21.7–50.3)	29 ± 5.1 (20.7–45.5)	^a^ 0.0002; ^c^ 0.0240 ^d^ 0.0007 ^e^ 0.0003 ^f^ 0.0041
**Comorbidities, n (%)**								
Hypertension	6 (10.9)	90 (44.1)	2 (5.5)	19 (31.7)	46 (68.6)	23 (56.1)	18 (23.4)	^a,d,e^ < 0.0001; ^c^ 0.0118
Cardiovascular disorder	7 (12.7)	21 (10.3)	4 (11.1)	9 (15)	6 (8.9)	2 (4.9)	-	
Diabetes mellitus	3 (5.4)	62 (30.4)	3 (8.3)	16 (32)	29 (43.3)	14 (34.1)	13 (16.9)	^a,d^ < 0.0001; ^c^ 0.0006 ^e^ 0.0004
History of smoking	6 (10.9)	39 (19.1)	4 (11.1)	9 (15)	15 (22.4)	11 (26.8)	2 (2.6)	
Neurological disorder	-	34 (16.7)	9 (25)	10 (16.7)	10 (14.9)	5 (12.2)	14 (18.2)	
**Presenting symptoms, n (%)**								
Dyspnea	-	127 (62.2)	-	45 (75)	47 (70.1)	35 (85.4)	60 (77.9)	^f^ 0.0157
Fever	-	64 (31.4)	2 (5.5)	14 (23.3)	33 (49.2)	15 (36.6)	53 (68.8)	^f^ < 0.0001
Myalgia	-	45 (22.1)	-	7 (11.7)	23 (34.3)	15 (36.6)	68 (88.3)	^f^ < 0.0001
Diarrhea	-	52 (25.5)	12 (33.3)	21 (35)	14 (20.9)	5 (12.2)	47 (61.1)	^f^ < 0.0001
Cough	-	145 (71.1)	26 (72.2)	42 (70)	51 (76.1)	26 (63.4)	53 (93)	^f^ 0.0004
Hyperactive delirium	-	12 (5.9)	-	5 (8.3)	-	7 (17.1)	-	
Dysgeusia	-	53 (26)	21 (58.3)	22 (36.7)	8 (12)	2 (4.9)	62 (80.5)	^f^ < 0.0001
Anosmia	-	58 (28.4)	22 (61.1)	23 (38.3)	11 (16.4)	2 (4.9)	58 (75.3)	^f^ < 0.0001
**Laboratory findings, M ± SD, and (IQR)**								
Erythrocytes × 10^9^/L	4.7 ± 0.5 (3.6–5.8)	4.5 ± 0.7 (2.2–5.9)	4.8 ± 0.5 (3.9–5.8)	4.5 ± 0.6 (3.0–5.9)	4.3 ± 0.8 (2.2–5.8)	4.0 ± 0.8 (2.3–5.7)	4.6 ± 0.4 (3.7–5.4)	^a^ 0.0076; ^d^ 0.0026; ^e^ < 0.0001
Hemoglobin (g/dL)	14.5 ± 1.5 (10.5–17.4)	13.3 ± 2.4 (6.6–18.2)	15 ± 1.2 (12–16.9)	13.6 ± 2.2 (8.1–18.2)	12.6 ± 2.3 (6.8–16.5)	12.4 ± 2.6 (6.6–18.2)	13.8 ± 1.4 (9.4–16.5)	^a,d,e^ < 0.0001; ^c^ 0.0142
Leukocytes × 10^9^/L	7.4 ± 1.8 (4.1–11.3)	8.4 ± 4.4 (1.6–26.1)	7.3 ± 2.3 (3.2–13.6)	7.4 ± 2.7 (2.6–15.7)	8.6 ± 4.1 (1.6–21.9)	11.1 ± 6.0 (4.6–26.1)	5.9 ± 1.8 (2.1–12.3)	^e^ < 0.0001; ^f^ 0.0098
Neutrophils × 10^9^/L	4.3 ± 1.3 (2.3–7.4)	6.0 ± 4.1 (1.6–23.8)	4.1 ± 1.7 (1.6–9.9)	5.0 ± 2.6 (1.6–13.4)	7.2 ± 3.5 (2.9–18.8)	9.5 ± 5.2 (3.2–23.7)	3.1 ± 1.3 (1.1–8.6)	^a,d,e^ < 0.0001; ^f^ 0.0299
Lymphocytes × 10^9^/L	2.3 ± 0.6 (1.0–3.9)	1.3 ± 0.9 (0.1–4.3)	2.3 ± 0.7 (1.1–4.3)	1.5 ± 0.8 (0.3–3.8)	1.0 ± 0.6 (0.1–2.8)	1.0 ± 0.5 (0.2–2.2)	2.1 ± 0.5 (1.0–3.6)	^a,d,e^ < 0.0001; ^c^ 0.0004
Neutrophil–lymphocyte ratio	1.9 ± 0.6 (1.0–3.3)	4.9 ± 5.6 (0.2–28.7)	1.7 ± 0.6 (0.7–3.6)	3.3 ± 3.1 (0.6–15.2)	6.8 ± 4.3 (1.0–23)	9.1 ± 6.7 (2.3–26.7)	1.5 ± 0.7 (0.5–4.3)	^a,d,e^ < 0.0001; ^c^ 0.0145
Monocytes × 10^9^/L	0.5 ± 0.1 (0.3–0.9)	0.5 ± 0.3 (0.1–1.6)	0.5 ± 0.1 (0.2–0.9)	0.4 ± 0.2 (0.1–1.1)	0.5 ± 0.3 (0.1–1.3)	0.5 ± 0.4 (0.1–1.6)	0.4 ± 0.1 (0.2–1.0)	
Platelets × 10^9^/L	212 ± 43.8 (129–363)	235 ± 89.5 (50–515)	233 ± 63.1 (135–365)	228 ± 93.8 (117–515)	257 ± 102 (85–506)	212 ±67 (50–370)	213 ± 54.6 (116–386)	
Glycemia (mg/dL)	89 ± 14.6 (63–146)	114.5 ± 69 (65–409)	87 ± 13.4 (71–127)	101 ± 33 (65–2003)	132 ± 78.4 (89–409)	143 ± 81 (79–384)	98.5± 18.6 (67–168)	^a,d,e^ < 0.0001; ^c^ 0.0109
**Hospital support, n (%)**								
Infirmary	-	100 (49.0)	-	34 (56.7)	63 (94)	3 (7.3)	-	
Intensive care unit	-	44 (21.6)	-	2 (3.3)	4 (6.0)	38 (92.7)	-	
**Hospitalization data, n**								
Days in hospital	-	9 ± 4.1 (1–19)	12 ± 4.9 (2–18)	9 ± 4.0 (1–19)	7 ± 3.2 (1–17)	9 ± 3.8 (4–19)	-	
Days from symptom onset to recruitment	-	4 ± 4.2 (1–17)	9 ± 3.7 (2–17)	4 ± 3.9 (1–15)	3 ± 3.5 (1–16)	3 ± 4.6 (1–16)	-	
Days recovery until recruitment	-	-	-	-	-	-	30 ± 17.4 (15–90)	
**Respiratory support received (%)**								
High flow nasal cannula	-	65 (31.9)	-	24 (40)	39 (58.2)	2 (4.8)	-	
Oxygen masks/noninvasive	-	35 (17.1)	-	3 (5)	26 (38.8)	6 (14.6)	-	
Invasive ventilation	-	33 (16.2)	-	-	1 (1.5)	32 (78)	-	
Oxygen saturation, M ± SD (IQR)	99 ± 1.8 (90–99)	94 ± 8.1 (54–99)	97.5 ± 1.7 (94–99)	96 ± 3.9 (80–99)	91 ± 8.6 (54–99)	89 ± 9.1 (60–96)	-	^a,d,e,f^ < 0.0001; ^c^ 0.0008
**Medications, n (%)**								
Glucocorticoid	2 (3.6)	125 (61.3)	5 (13.9)	30 (50)	55 (82.1)	35 (85.4)	-	^a^ < 0.0001
Azithromycin	-	121 (59.3)	8 (22.2)	39 (65)	46 (68.6)	28 (68.3)	-	
Ceftriaxone	-	93 (45.6)	-	23 (38.3)	46 (68.7)	24 (58.5)	-	
Oseltamivir	-	60 (29.4)	4 (11.1)	10 (16.7)	34 (50.7)	12 (29.3)	-	
Colchicine	-	6 (2.9)	-	1 (1.7)	-	5 (12.2)	-	
CQ/HCQs	-	27 (13.2)	-	4 (6.7)	13 (19.4)	10 (24.4)	-	
Anticoagulant	-	18 (8.8)	1 (2.8)	7 (11.7)	1 (1.5)	9 (21.9)	-	
Ivermectin	-	11 (5.4)	5 (13.9)	6 (10)	-	-	-	

Data were median (min-max) or n (%). Patient data were compared using the chi-square test or Fisher’s exact test for categorical variables and one-way analysis of variance (ANOVA) Mann–Whitney, non-parametric *t*-test was used for continuous variables; *p* < 0.05 was considered statistically significant. Abbreviations: SD: standard deviation. Data are median interquartile range (IQR), n (%), or n/N. ^a^ Comparisons between the healthy controls and all COVID-19 patients; ^b^ Healthy controls and Asy-to-mild group; ^c^ Healthy controls and Moderate group; ^d^ Healthy controls and Severe group; ^e^ Healthy controls and Critical group. ^f^ Healthy controls and Convalescent group.

## Data Availability

Normalized transcriptomic data from the participants’ whole-blood leukocytes and associated metadata were deposited on the ArrayExpress database at EMBL-EBI (www.ebi.ac.uk/arrayexpress accessed on 22 June 2022) under accession number E-MTAB-11240.

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
