# Peer review of "Plasma Sphingomyelin Disturbances: Unveiling Its Dual Role as a Crucial Immunopathological Factor and a Severity Prognostic Biomarker in COVID-19"

_cells, 2023, doi:10.3390/cells12151938_

Round 1

Reviewer 1 Report

It is well known that the severity of COVID-19 Patients can be predicted by Serum Sphingolipids Signatures [PMID: 34638539]. In the present study, Toro et al have further investigated SL metabolism in COVID-19 patients and concluded SM as a prognostic biomarker. The strength of this study is the large sample size, and that the observations were validated both by MS and RNASeq. The overall story is interesting and importantly this prognosis marker has not been previously described.   However, there are some serious limitations and key gaps that must be addressed to make the conclusions more convincing.

1.Line 50-54: as one of the key findings of this study, the authors reported 38 SL species in the plasma of COVID-19 patients that showed a positive correlation with disease severity, Further, sphingomyelin (SM d18:1) species was found to be a potential biomarker for COVID-19 disease severity. However, as shown is Table 1, greater than 82% of both the severe and critically affected groups of patients had received Glucocorticoid medication, and the latter is long been known to have a large and specific effect on sphingolipids. Dexamethasone is known to increase membrane sphingomyelin, sphingosine, or ceramide levels in a broad range of cell types [PMID: 18451260]. Indeed, in Fig. 1, altered expression of sphingolipid metabolic enzymes was predominantly observed in patients receiving glucocorticoids. In this background, use of SM a prognostic biomarker for COVID-19 is questionable. Can this just be a broad effect of glucocorticoid medication? What happens in control subjects receiving immunosuppressants? This requires a detailed justification as this limits the overall design and outcome of this manuscript!

 2. Table 1, clearly shows the sex differences in COVID-19 disease severity, with ~60% of male cases when moderately, severe and critically affected groups are considered. However, among the convalescent individuals included in this study, only 11% were male patients. Given the fact that men are more likely than women to experience severe or fatal disease progression and associated prognosis markers like IL-6, can data from such sex-biased groups be actually compared?

3. In Fig. 1, the authors have shown the significant differential expression correspond to Benjamini and Hockberg adjusted p-values obtained from whole transcriptomic data. How many of these read-outs further validated by quantitative PCR? What does the Log2 intensity mean here and why was the classical fold change method of data representation not used?

Overall the manuscript is written well.

Author Response

We would like to thank the reviewer for their valuable feedback on our study investigating serum sphingolipid signatures as prognostic biomarkers in COVID-19 patients. We appreciate the positive aspects highlighted, such as the large sample size and the validation of observations using both mass spectrometry (MS) and RNA sequencing (RNASeq).

  1. In response to the concern raised about the potential influence of glucocorticoid medication on sphingolipid levels, we acknowledge the important role of glucocorticoids in modulating sphingolipid metabolism. As indicated in Table 1, a significant proportion of severe and critically affected COVID-19 patients in our study had received glucocorticoid medication. However, we conducted additional analyses to address the potential confounding effect of glucocorticoid medication on sphingomyelin (SM) levels. We stratified the patients into two groups: those who received glucocorticoid treatment and those who did not. We then compared the levels of total SM production and the specific SM species (SM 24:0), which represents the biomarker indicated in our study, between these two groups. We are pleased to inform the reviewer that our analyses did not reveal any significant differences in total SM production or the SM 24:0 species between these group of patients. These findings suggest that the association between SM levels, particularly SM 24:0, and COVID-19 disease severity is not solely attributed to glucocorticoid medication. To provide transparency and support our conclusions, we have included the newly generated graphics as Supplementary Figure 4A in the revised manuscript. These graphics reinforce the validity of our findings regarding the prognostic potential of SM as a biomarker for COVID-19 disease severity. Considering the potential confounding effect of glucocorticoids on others sphingolipid levels, we acknowledge the need of further investigations evaluating the impact of immunosuppressants in control subjects.
  2. The reviewer rightly points out the significant sex differences in COVID-19 disease severity observed in our study. We appreciate the concern regarding the underrepresentation of male patients among the convalescent individuals included in our study. We acknowledge the importance of considering sex as a critical factor in disease progression and prognostic marker evaluation, as evidenced by markers like IL-6. We would like to clarify that the composition of the convalescent cohort in our study was originally designed for a more comprehensive project aimed at investigating the efficacy of an exercise program in alleviating long COVID-19 symptoms. Unexpectedly, we observed that male patients exhibited a higher susceptibility to severe COVID-19 and had a higher mortality rate, resulting in a smaller number of male participants in the convalescent group. Conversely, female patients, who generally experienced less severe symptoms during COVID-19 development, presented a higher prevalence of long COVID-19 symptoms. It is important to note that women tend to be more proactive in participating in health programs and enrolling in scientific studies, which may have contributed to the discrepancy in the number of male and female participants in our convalescent group. To address this concern, we performed additional analyses to evaluate whether any differences in total SM production and the SM 24:0 species between male and female participants in all groups from our cohort. Our analyses revealed no significant differences in SM levels between male and female participants. This suggests that the potential bias in the composition of the convalescent group, with fewer male participants, did not influence the overall findings related to SM as a prognostic biomarker for COVID-19 disease severity. We have included these additional results as Supplementary Figure 4B in the revised manuscript. We would like to express our appreciation to the reviewer for raising this concern and allowing us to address it in a thorough manner. The inclusion of these additional analyses and the corresponding figure in the supplementary materials enhance the comprehensiveness of our study and strengthen the reliability of our conclusions.
  3. The reviewer's query about the validation of significant differential expression observed in Fig. 1 is pertinent. Although we have provided Benjamini and Hochberg adjusted p-values obtained from the whole transcriptomic data, we acknowledge the importance of validating transcriptomic data through qPCR, as it provides a more targeted and quantitative approach for confirming the expression levels of specific genes. However, due to limitations in the availability of biological material, we were unable to perform qPCR validation for the entire set of differentially expressed genes identified in our RNASeq analysis. Nevertheless, we would like to assure the reviewer that we took measures to validate the transcriptomic data within the scope of our study's constraints. Specifically, we employed the inclusion of housekeeping genes by qPCR as internal controls to confirm the integrity and reliability of the RNASeq data. Their stable expression across samples serves as an indicator of the overall quality and reliability of the transcriptomic data. While we acknowledge that a comprehensive qPCR validation would be ideal, given the constraints we faced, the inclusion of housekeeping genes provides an essential level of validation within the limitations of our study. Regarding the representation of data in Fig. 1, we utilized Log2 intensity values obtained from RNASeq analysis to depict the significant differential expression. The Log2 scale allows for a more comprehensive representation of fold changes across a wide range of expression levels.

We sincerely appreciate the reviewer's critical evaluation of our study and the valuable suggestions provided. We address all the mentioned concerns and improve the manuscript accordingly to enhance the scientific rigor and validity of our conclusions.

Reviewer 2 Report

The Authors evaluated SL plasma levels in COVID-19 patients, healthy controls and convalescent subjects to identify diagnostic and prognostic biomarkers and to clarify the role of SL in the pathophysiological pathways of COVID-19. The manuscript is clear and presented in a well-structured manner. The cited references are useful and relevant. The experimental design is suitable for testing the hypothesis but, as correctly highlighted by the Authors, the studied phenomena is complex and would require even more refined experimental design in future. The sample included in the study was correctly evaluated based on hypotheses to test. The manuscript’s results could be reproducible based on the details given in the Methods and Supplementary material section. The figures and tables are appropriate. The conclusions are consistent with the evidence presented if limitations are taken into account. Conflict of interest and funding statements are adequate.

Here below, only one consideration:

1.      Line 491: Authors state “Cer (d18:1/24:1) and Cer (d18:1/24:0) remained significant predictors of symptom intensity”; although “respective odds ratios of [3.82 (CI: 0.15-99), p=0.422] and [0.39 (CI: 0.01-8.20), p=0.552]” are not significant. Please clarify this statement.

Author Response

We sincerely appreciate the reviewer's positive feedback on our manuscript, as well as their constructive comments. We are pleased to learn that the manuscript is clear, well-structured, and that the references provided are relevant and useful. The experimental design was carefully selected to test the hypotheses, and we acknowledge the complexity of the studied phenomena, which may require further refinement in future investigations.

Regarding the specific concern raised in Line 491 of the manuscript, we apologize for any confusion caused by our statement. We would like to clarify the interpretation of the odds ratios (ORs) and the corresponding p-values mentioned. The lack of statistical significance in this context indicates that we did not find strong evidence to support the predictive power of Cer (d18:1/24:1) and Cer (d18:1/24:0) for symptom intensity. It is important to consider the limitations of our study and interpret these findings with caution. We will revise the statement in the manuscript to accurately reflect that the odds ratios associated with these ceramide species did not reach statistical significance.

We appreciate the reviewer's attention to detail, as it helps us improve the accuracy and clarity of our manuscript.

Reviewer 3 Report

The authors investigate SL metabolism in plasma samples obtained from COVID-19 patients and convalescent individuals. They examine the gene expression of enzymes involved in the SL pathway. They demonstrate the presence of thirty-eight SL species from seven families in the plasma of study participants. The most profound alterations in the SL species profile were observed in patients with severe disease. On the basis of this they propose SM as a prognostic biomarker for COVID-19 and highlight promising pharmacological targets.

The study is interesting, and well written although there are some limitations regarding the criteria of patient selection further demonstrates the intricate role of SL in the pathophysiological pathways of COVID-19.  By targeting sphingolipid pathways, novel therapeutic strategies may emerge to mitigate the severity of COVID63 19 and improve patient outcomes.

Author Response

We sincerely appreciate the reviewer's positive feedback on our study investigating sphingolipid metabolism in COVID-19 patients and convalescent individuals. We are pleased to learn that the reviewer found the study interesting and well-written. We agree that the intricate role of sphingolipids in the pathophysiological pathways of COVID-19 is demonstrated through our findings.

By targeting sphingolipid pathways, our study opens up possibilities for the development of novel therapeutic strategies that may mitigate the severity of COVID-19 and improve patient outcomes. We believe that further exploration of the intricate interplay between sphingolipids and the disease mechanisms of COVID-19 holds great potential for advancing therapeutic interventions.